# Prognostic value of postoperative decrease in serum albumin on surgically resected early-stage non-small cell lung carcinoma: A multicenter retrospective study

**Fumihiko Kinoshita**[1], **Tetsuzo Tagawa**[1]\*, **Takanori Yamashita**[2], **Tomoyoshi Takenaka**[1], **Taichi Matsubara**[3], **Gouji Toyokawa**[4], **Kazuki Takada**[1], **Taro Oba**[1], **Atsushi Osoegawa**[1], **Koji Yamazaki**[4], **Mitsuhiro Takenoyama**[3], **Mototsugu Shimokawa**[5], **Naoki Nakashima**[2], **Masaki Mori**[1]

1 Department of Surgery and Science, Graduate School of Medical Sciences, Kyushu University, Fukuoka, Japan, 2 Medical Information Center, Kyushu University Hospital, Fukuoka, Japan, 3 Department of Thoracic Oncology, National Hospital Organization Kyushu Cancer Center, Fukuoka, Japan, 4 Department of Thoracic Surgery, Clinical Research Institute, National Hospital Organization, Kyushu Medical Center, Fukuoka, Japan, 5 Department of Biostatistics, Graduate School of Medicine, Yamaguchi University, Yamaguchi, Japan

\* t_tagawa@surg2.med.kyushu-u.ac.jp

**Data Availability Statement:** Sharing of de-identified data sets is restricted by the Kyushu University Research Ethics Review Committee

## Abstract

### Background

Preoperative nutritional status is an important host-related prognostic factor for non-small cell lung carcinoma (NSCLC); however, the significance of postoperative changes in nutritional status remains unclear. This study aimed to elucidate the significance of postoperative decreases in serum albumin (ΔAlb) on the outcomes of early-stage NSCLC.

### Methods

We analyzed 443 training cohort (TC) and 642 validation cohort (VC) patients with pStage IA NSCLC who underwent surgery and did not recur within 1 year. We measured preoperative serum albumin levels (preAlb) and postoperative levels 1 year after surgery (postAlb), and calculated ΔAlb as (preAlb − postAlb)/preAlb × 100%. A cutoff value of 11% for ΔAlb was defined on the basis of the receiver operating characteristic curve for the TC.

### Results

Patients were divided into ΔAlb-Decreased and ΔAlb-Stable groups, including 100 (22.6%) and 343 (77.4%) in the TC, and 58 (9.0%) and 584 (90.1%) in the VC. ΔAlb-Decreased was associated with male sex ($p = 0.0490$), smoking ($p = 0.0156$), and non-adenocarcinoma ($p<0.0001$) in the TC, and pT1b ($p = 0.0169$) and non-adenocarcinoma ($p = 0.0251$) in the VC. Multivariable analysis identified ΔAlb as an independent prognostic factor for disease-free survival (DFS) and overall survival (OS) in both cohorts (VC: DFS, HR = 1.9, 95%CI: 1.10–3.15, $p = 0.0197$; OS, HR = 2.0, 95%CI: 1.13–3.45, $p = 0.0173$). Moreover, subgroup

because they contain patient information that may be identified. Please contact the Kyushu University Research Ethics Review Board or the Corresponding Author when sending a data request. Kyushu University Research Ethics Review Committee: 3-1-1, Maidashi, Higashi-ku, Fukuoka, 812-8582, Japan TEL: 092-642-5774 FAX: 092-642-5775 E-MAIL: ijkseimei@jimu. kyushu-u.ac.jp.

**Funding:** The funders had no role in study design, data collection and analysis, decision to publish, or preparation of the manuscript.

**Competing interests:** The authors have declared that no competing interests exist.

analysis demonstrated that the prognostic value of ΔAlb was consistent for age, sex, smoking history, surgical procedure, and histological type.

## Conclusion

We demonstrated a negative impact of postoperative decrease of the serum albumin on the prognosis of patients with early-stage NSCLC. Postoperative changes in nutritional status might be important in NSCLC outcomes.

## Introduction

The preoperative nutritional status of patients, reflected by factors such as serum albumin [1], Controlling Nutritional Status score [2,3], geriatric nutritional risk index [4], prognostic nutritional index [5,6], Glasgow prognostic score [7], body mass index [8,9], and skeletal muscle area (SMA) [10–13], has attracted much attention as an important host-related prognostic factor in patients with non-small cell lung carcinoma (NSCLC).

A previous study showed that a postoperative decrease in serum albumin (ΔAlb) 1 day after surgery was associated with postoperative pulmonary complications in patients with NSCLC [14]. However, the prognostic significance of ΔAlb on the long-term outcomes of NSCLC patients after surgery remains unclear.

We previously showed that a decrease in SMA 1 year after surgery was associated with a poor prognosis of NSCLC [12,13]. Furthermore, a decrease in SMA correlated with exacerbation of nutritional indices, and postoperative deterioration of nutritional status was related to a poor prognosis of NSCLC [13].

In this study, we analyzed the clinical significance of ΔAlb at 1 year after surgery, as a convenient marker of nutritional status, on the long-term outcomes of patients with pathological stage (pStage) IA NSCLC.

## Patients and methods

### Study patients

This study was reviewed and approved by our institutional review boards (Kyushu University Hospital [KUH], IRB No. 2019–232; Kyushu Cancer Center [KCC], IRB No. 2019–56; Kyushu Medical Center [KMC], IRB No. 19D152). Regarding participant consent, the informed consent for this research was waived by the ethics committee.

This study included 1210 Japanese patients with pStage IA NSCLC who underwent surgical resection between January 2003 and December 2014 at KUH (545 patients) and between January 2009 and December 2014 at KCC (324 patients) and KMC (341 patients). Of these, 59 patients who recurred within 1 year after surgery were excluded (KUH, 44 patients; KCC, six patients; KMC, nine patients). Furthermore, because of incomplete resection, pretreatment before surgery, or lack of data, 66 patients were also excluded (KUH, 58 patients; KCC, three patients; KMC, five patients). Patients who received chemotherapy within 1 year after surgery including adjuvant chemotherapy were also excluded from this study. Finally, 1085 patients were enrolled in this study (KUH, 443 patients; KCC, 315 patients; KMC, 327 patients). We first examined patients from KUH as a training cohort (TC). Subsequently, we validated our findings in an independent validation cohort (VC) composed of patients from KCC and

KMC. A total of 443 and 642 patients were enrolled in the TC and VC, respectively. The consort diagram of patient selection was shown in **Fig 1**.

Clinicopathological characteristics, disease-free survival (DFS), and overall survival (OS) were analyzed retrospectively. Clinicopathological characteristics included age, sex, smoking history, pulmonary comorbidities, surgical procedure, pathological T status (pT), histological type, vascular invasion, and lymphatic invasion. Pulmonary comorbidities included interstitial pneumonia, chronic obstructive pulmonary disease, or asthma. The pStage was defined according to the 7th edition of TNM classification [15]. The clinical information and follow-up data were obtained from the patients' medical records.

### Evaluation of serum albumin

Preoperative serum albumin levels (preAlb) were measured before surgery and postoperative serum albumin levels (postAlb) at the closest point to 12 months at 9–15 months after surgery. For measuring serum albumin levels, all of three institutions used the modified bromocresol purple method. ΔAlb was calculated as (preAlb − postAlb)/preAlb × 100%, as in previous reports [14,16,17].

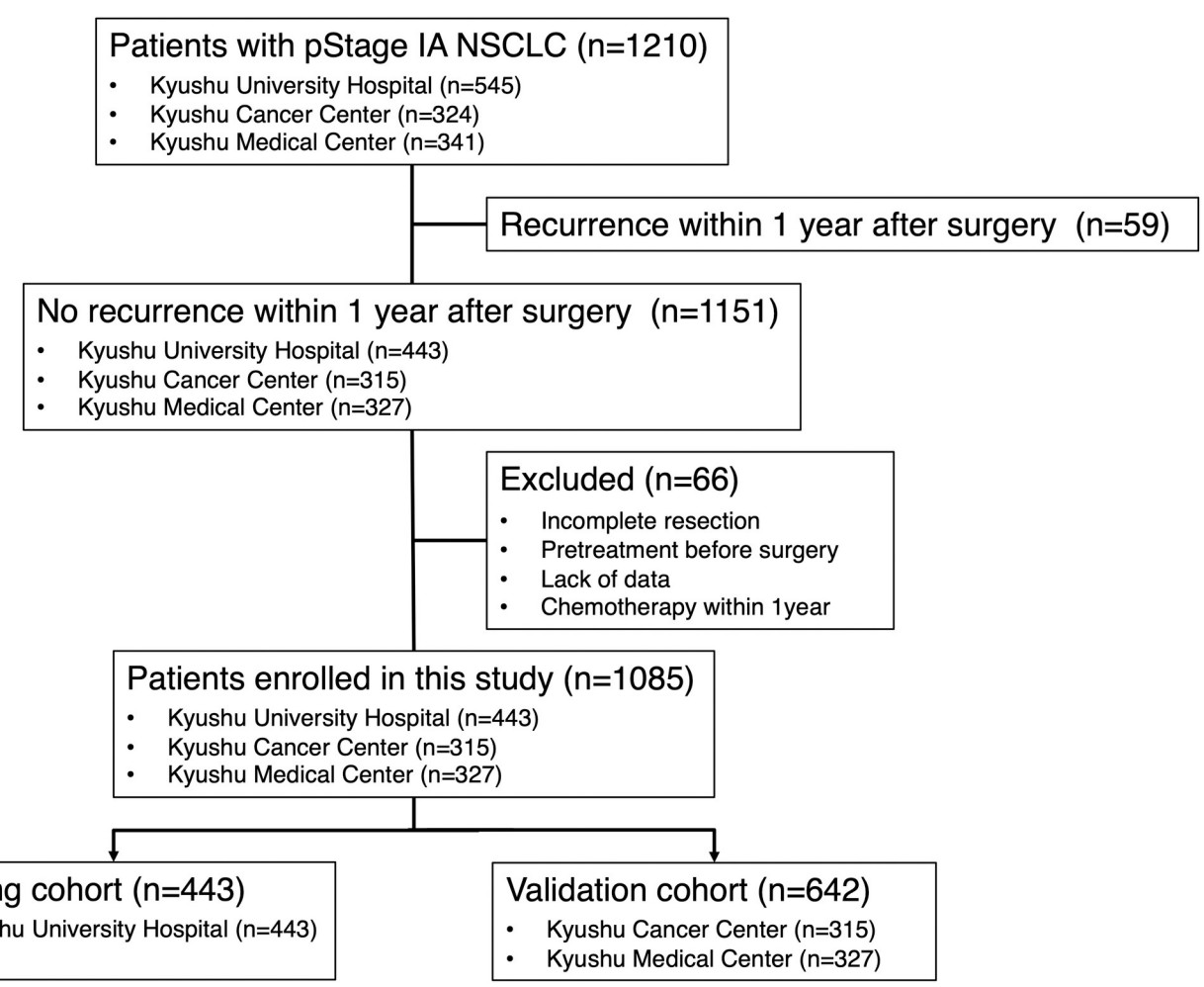

**Fig 1. Consort diagram of patient selection.** pStage, pathological stage; NSCLC, non-small cell lung carcinoma.

## Statistical analysis

All statistical analyses were performed using JMP pro 14.0 software (SAS Institute). Patient characteristics were analyzed by Pearson's $\chi^2$ test. DFS was defined as the time between surgery and the date of last follow-up, recurrence, or death; OS was the time between surgery and the date of last follow-up or death. Survival curves were estimated by the Kaplan–Meier method with the log-rank test. Hazard ratios for positive risk factors were estimated by Cox proportional hazards regression analysis with backwards elimination. As for multivariable analysis, after applying multivariable analysis to all factors used in univariable analysis at the same time, we eliminated less important factors with high p-values one by one and reduced factors so that only factors with $p < 0.05$ were included. All results were considered statistically significant at $p < 0.05$.

## Results

### Patient characteristics

A total of 1085 patients with pStage IA NSCLC who underwent complete surgical resection and did not recur within 1 year were enrolled in this study. We classified 443 patients from KUH as the TC, and 642 patients from KCC and KMC as the VC. The detailed clinicopathological characteristics of the two cohorts are shown in **S1 Table**. In the TC, 214 (48.3%), 223 (50.3%), and 232 (52.4%) patients were of older age ($\geq$70 years), male, and smoked, respectively. The median follow-up period was 5.47 years (range, 1.01–15.97 years). In terms of the VC, 279 (43.5%), 283 (44.1%), and 295 (46.0%) patients were of older age ($\geq$70 years), male, and smoked, respectively, and the median follow-up period was 5.75 years (range, 1.36–10.59 years).

In terms of serum albumin, the median preAlb and ΔAlb in the TC were 4.3 g/dL (range, 3.1 to 5.3 g/dL) and 4.8% (range, -29.2 to 79.2%), respectively. In the VC, the median preAlb and ΔAlb in the TC were 4.3 g/dL (range, 3.0 to 5.4 g/dL) and 2.4% (range, -30.0 to 43.9%), respectively. The histograms of distributions of preAlb and ΔAlb in each cohort were shown in **S1 Fig**.

### Clinicopathological characteristics associated with ΔAlb

The cutoff value of ΔAlb was set as 11% based on the receiver operating characteristic curve (ROC) curve for 5-year OS in the TC (**Fig 2**). One hundred (22.6%) patients in the TC and 58 (9.0%) patients in the VC were classified as ΔAlb-Decreased (ΔAlb $\geq$11%), respectively. The association between clinicopathological characteristics and ΔAlb is shown in **Table 1**. ΔAlb-Decreased in the TC was significantly associated with male sex ($p = 0.0490$), smoking ($p = 0.0156$), and non-adenocarcinoma ($p < 0.0001$). In the VC, ΔAlb-Decreased was significantly associated with pT1b ($p = 0.0169$) and non-adenocarcinoma ($p = 0.0251$).

### Survival analysis according to ΔAlb

We evaluated the difference in survival between the ΔAlb-Decreased and ΔAlb-Stable groups by Kaplan–Meier analysis. DFS and OS of the ΔAlb-Decreased group in the TC were significantly poorer compared with the ΔAlb-Stable group (5-year DFS, 77.2% versus 94.8%, $p < 0.0001$; 5-year OS, 84.1% versus 98.2%, $p < 0.0001$) (**Fig 3A and 3B**). Furthermore, in the VC, DFS and OS were also significantly worse in the ΔAlb-Decreased group compared with the ΔAlb-Stable group (5-year DFS, 78.0% versus 87.7%, $p = 0.0015$; 5-year OS, 83.4% versus 90.6%, $p = 0.0010$) (**Fig 3C and 3D**).

### Survival analysis according to preAlb

The cutoff value of preAlb was set as 4.1 g/dL based on the ROC curve for 5-year OS in the TC (**S2 Fig**). The clinicopathological characteristics according to preAlb are shown in **S2**

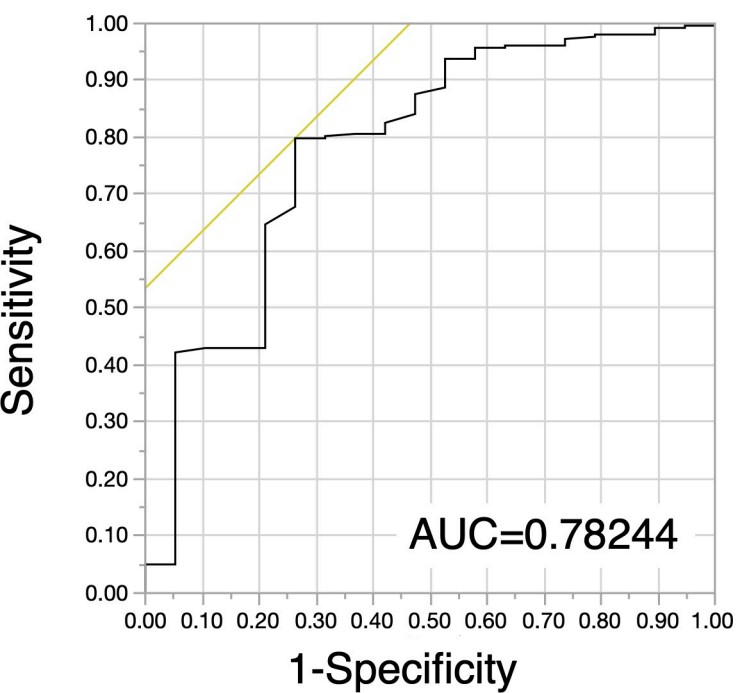

**Fig 2. Receiver operating characteristic curve (ROC) of training cohort (TC).** ROC curves for postoperative decrease in serum albumin (ΔAlb) in the TC. AUC: Area under the curve.

Table, and preAlb-Low was significantly associated with poor prognosis in both the TC and VC (S3 Fig).

## Univariable and multivariable analyses of survival

We performed univariable and multivariable analyses of survival. In the TC, multivariable analysis identified smoker ($p$ = 0.0233), lobectomy ($p$ = 0.0175), adenocarcinoma ($p$ = 0.0187), lymphatic invasion ($p$ = 0.0180), and ΔAlb-Decreased ($p$ = 0.0003) as independent prognostic factors for DFS, and lobectomy ($p$ = 0.0015), adenocarcinoma ($p$ = 0.0004), ΔAlb-Decreased ($p$<0.0001), and preAlb-Low ($p$ = 0.0010) as independent factors for OS (S3 Table). In the VC, multivariable analysis identified older age ($p$<0.0001), male sex ($p$<0.0001), pulmonary comorbidity ($p$ = 0.0017), and ΔAlb-Decreased ($p$ = 0.0186) as independent prognostic factors for DFS, and older age ($p$<0.0001), male sex ($p$ = 0.0001), pulmonary comorbidity ($p$ = 0.0006), lobectomy ($p$ = 0.0247), and ΔAlb-Decreased ($p$ = 0.0145) as independent prognostic factors for OS (Table 2).

Furthermore, a forest plot of the subgroup analysis demonstrated that the prognostic value of ΔAlb was almost consistent across different subgroups in both the TC (S4 Fig) and VC (Fig 4).

## Combined survival analysis of ΔAlb and preAlb

To elucidate the prognostic significance of ΔAlb in more detail, we performed a combined analysis of ΔAlb and preAlb. Patients were categorized into the following four groups: Alb-Low-Decreased (preAlb <4.1g/dL and ΔAlb ≥11%), Alb-Low-Stable (preAlb <4.1g/dL and ΔAlb <11%), Alb-High-Decreased (preAlb ≥4.1g/dL and ΔAlb ≥11%), and Alb-High-Stable (preAlb ≥4.1g/dL and ΔAlb <11%). DFS and OS differed significantly among the four groups

**Table 1. Associations of clinicopathological characteristics with ΔAlb.**

| Characteristics | Training cohort | | | | | | Validation cohort | | | | | |
|---|---|---|---|---|---|---|---|---|---|---|---|---|
| | ΔAlb | | | | | | ΔAlb | | | | | |
| | Decreased (n = 100) | | Stable (n = 343) | | *p* value | | Decreased (n = 58) | | Stable (n = 584) | | *p* value | | |
| Age, years | | | | | | | | | | | | |
| <70 | 50 | (50.0%) | 179 | (52.2%) | 0.7002 | | 26 | (44.8%) | 337 | (57.7%) | 0.0592 | |
| ≥70 | 50 | (50.0%) | 164 | (47.8%) | | | 32 | (55.2%) | 247 | (42.3%) | | |
| Sex | | | | | | | | | | | | |
| Female | 41 | (41.0%) | 179 | (52.2%) | 0.0490 | | 26 | (44.8%) | 333 | (57.0%) | 0.0744 | |
| Male | 59 | (59.0%) | 164 | (47.8%) | | | 32 | (55.2%) | 251 | (43.0%) | | |
| Smoking | | | | | | | | | | | | |
| Never smoker | 37 | (37.0%) | 174 | (50.7%) | 0.0156 | | 27 | (46.6%) | 320 | (54.8%) | 0.2296 | |
| Smoker | 63 | (63.0%) | 169 | (49.3%) | | | 31 | (53.4%) | 264 | (45.2%) | | |
| Pulmonary comorbidity | | | | | | | | | | | | |
| Absent | 93 | (93.0%) | 332 | (96.8%) | 0.1440 | | 53 | (91.4%) | 552 | (94.5%) | 0.3678 | |
| Present | 7 | (7.0%) | 11 | (3.2%) | | | 5 | (8.6%) | 32 | (5.5%) | | |
| Surgical procedure | | | | | | | | | | | | |
| ≥Lobectomy | 64 | (64.0%) | 221 | (64.4%) | 0.9368 | | 41 | (70.7%) | 416 | (71.2%) | 0.9306 | |
| Sublobar resection | 36 | (36.0%) | 122 | (35.6%) | | | 17 | (29.3%) | 168 | (28.8%) | | |
| pT | | | | | | | | | | | | |
| T1a | 73 | (73.0%) | 256 | (74.6%) | 0.7420 | | 32 | (55.2%) | 411 | (70.4%) | 0.0169 | |
| T1b | 27 | (27.0%) | 87 | (25.4%) | | | 26 | (44.8%) | 173 | (29.6%) | | |
| Histological type | | | | | | | | | | | | |
| Adenocarcinoma | 74 | (74.0%) | 310 | (90.4%) | <0.0001 | | 46 | (79.3%) | 525 | (89.9%) | 0.0251 | |
| Non-adenocarcinoma | 26 | (26.0%) | 33 | (9.6%) | | | 12 | (20.7%) | 59 | (10.1%) | | |
| Vascular invasion | | | | | | | | | | | | |
| Negative | 87 | (87.0%) | 315 | (91.8%) | 0.1686 | | 57 | (98.3%) | 568 | (97.3%) | 0.6459 | |
| Positive | 13 | (13.0%) | 28 | (8.2%) | | | 1 | (1.7%) | 16 | (2.7%) | | |
| Lymphatic invasion | | | | | | | | | | | | |
| Negative | 96 | (96.0%) | 337 | (98.3%) | 0.1824 | | 53 | (91.4%) | 546 | (93.5%) | 0.5391 | |
| Positive | 4 | (4.0%) | 6 | (1.7%) | | | 5 | (8.6%) | 38 | (6.5%) | | |

ΔAlb, postoperative decrease in serum albumin; pT, pathological T status.

in both the TC (*p*<0.0001 and *p*<0.0001, respectively) and VC (*p* = 0.0006 and *p* = 0.0001, respectively) (**Fig 5A and 5B**). The 5-year DFS rates in the Alb-Low-Decreased, Alb-Low-Stable, Alb-High-Decreased, and Alb-High-Stable groups were 48.3%, 92.8%, 82.4%, and 95.3% in the TC, respectively, and 68.6%, 81.4%, 79.2%, and 89.2% in the VC, respectively (**Fig 5C and 5D**). The 5-year OS rates were 54.3%, 93.9%, 89.3%, and 99.4% in the TC, respectively, and 68.6%, 86.0%, 85.3%, and 91.5% in the VC, respectively. The Alb-Low-Decreased group had the worst prognosis for DFS and OS; however, DFS and OS of the Alb-Low-Stable group were not significantly worse compared with the Alb-High-Decreased group and the Alb-High-Stable group in both cohorts.

## Discussion

The present study demonstrated the prognostic impact of ΔAlb on the long-term outcomes of patients with pStage IA NSCLC. Moreover, our subgroup analysis demonstrated that the prognostic value of ΔAlb was consistent for age, sex, smoking history, surgical procedure, and histological type. In addition, combined survival analysis of ΔAlb and preAlb suggested that, even

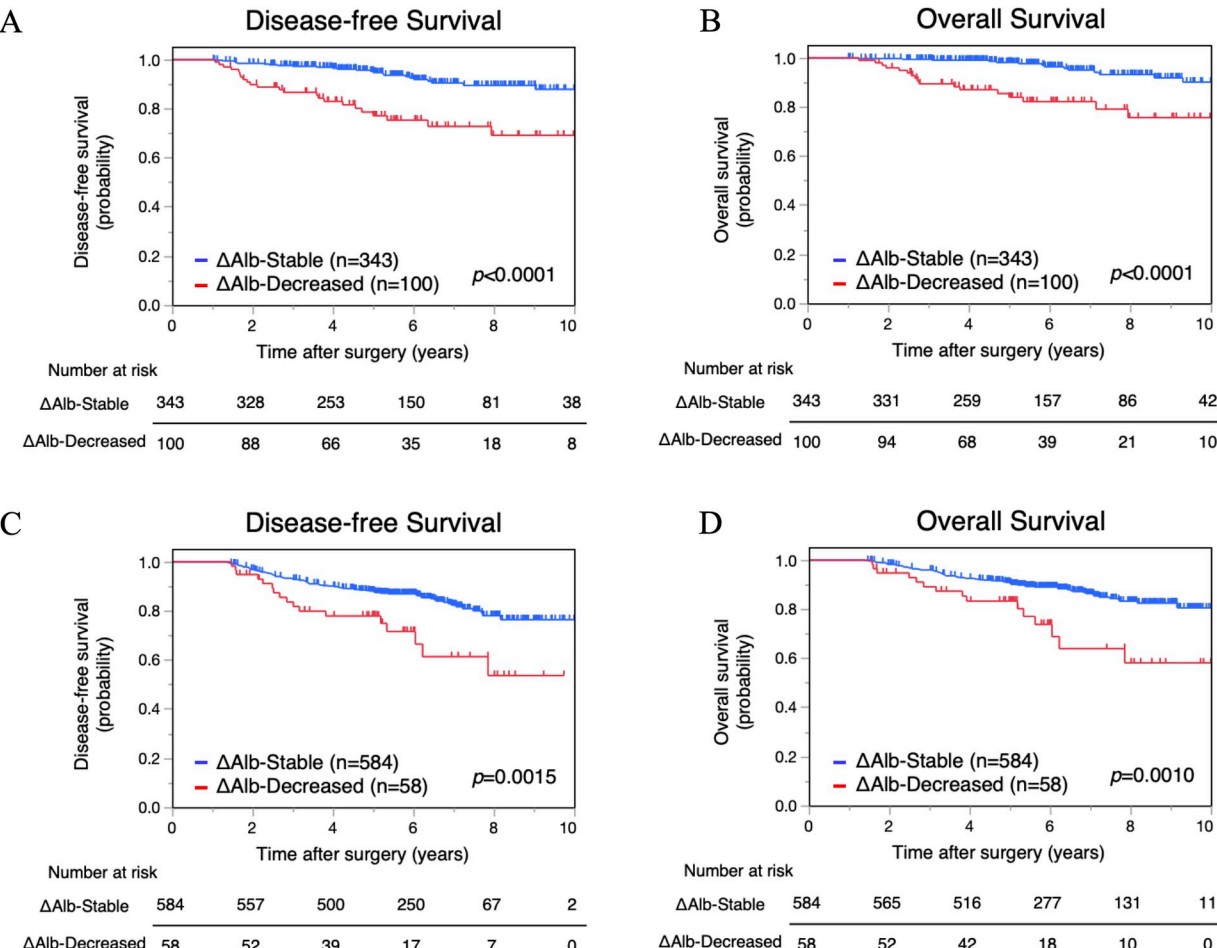

**Fig 3. Survival analysis according to postoperative decrease in serum albumin (ΔAlb).** Disease-free survival and overall survival of the ΔAlb-Decreased and ΔAlb-Stable groups in the training cohort (A, B) and in the validation cohort (C, D).

if nutritional status was poor before surgery, patients with maintained nutritional status had a better prognosis.

Serum albumin is one of the most convenient and important indicators for assessing nutritional status, and previous reports have identified preAlb as an essential prognostic factor for several cancers, including NSCLC [1,18]. Nutritional status is an important determinant of the immune response, and decreased albumin levels may be associated with an impaired antitumor immune response [19,20]. In addition, low levels of serum albumin reflect systemic inflammation, which is significantly associated with cancer progression [21–23]. Serum albumin is also an endogenous antioxidant [24,25]. Although the association between endogenous antioxidants and cancer progression remains unclear, a previous study demonstrated that loss of albumin resulted in a plasma redox imbalance and promoted cancer growth and metastasis [25]. These findings may help to explain the association between serum albumin and the prognosis of NSCLC.

We previously showed the prognostic significance of postoperative changes in nutritional status in patients with NSCLC by analyzing the postoperative decrease in SMA 1 year after surgery [12,13]. Despite its usefulness, evaluation of SMA requires complicated analysis of computed tomography results. In contrast, serum albumin can be conveniently measured by

**Table 2. Univariable and multivariable analyses of disease-free survival and overall survival in the validation cohort.**

| Characteristics | | \multicolumn{6}{c}{Disease-free survival} | | | | | | \multicolumn{6}{c}{Overall survival} | | | | | |
|---|---|---|---|---|---|---|---|---|---|---|---|---|---|---|
| | | \multicolumn{3}{c}{Univariable analysis} | \multicolumn{3}{c}{Multivariable analysis} | \multicolumn{3}{c}{Univariable analysis} | \multicolumn{3}{c}{Multivariable analysis} | | |
| | | HR | 95% CI | p value | HR | 95% CI | p value | HR | 95% CI | p value | HR | 95% CI | p value |
| Age | ≥70 | 2.3 | 1.53–3.34 | <0.0001 | 2.4 | 1.59–3.54 | <0.0001 | 2.9 | 1.87–4.52 | <0.0001 | 3.0 | 1.93–4.74 | <0.0001 |
| Sex | Male | 2.6 | 1.74–3.92 | <0.0001 | 2.4 | 1.61–3.64 | <0.0001 | 2.8 | 1.79–4.37 | <0.0001 | 2.7 | 1.70–4.21 | <0.0001 |
| Smoking | Smoker | 2.2 | 1.48–3.29 | 0.0001 | | | | 2.2 | 1.44–3.43 | 0.0003 | | | |
| Pulmonary comorbidity | Present | 2.6 | 1.48–4.73 | 0.0010 | 2.6 | 1.43–4.73 | 0.0017 | 3.0 | 1.64–5.55 | 0.0004 | 3.0 | 1.60–5.64 | 0.0006 |
| Surgical procedure | ≥Lobectomy | 0.7 | 0.47–1.04 | 0.0793 | | | | 0.6 | 0.41–0.97 | 0.0355 | 0.6 | 0.40–0.94 | 0.0247 |
| pT | T1b | 1.6 | 1.10–2.39 | 0.0152 | | | | 1.6 | 1.05–2.45 | 0.0288 | | | |
| Histological type | Adenocarcinoma | 0.4 | 0.24–0.61 | <0.0001 | | | | 0.3 | 0.21–0.56 | <0.0001 | | | |
| Vascular invasion | Positive | 1.9 | 0.69–5.10 | 0.2175 | | | | 2.2 | 0.81–6.05 | 0.1198 | | | |
| Lymphatic invasion | Positive | 1.8 | 0.92–3.37 | 0.0911 | | | | 1.7 | 0.81–3.48 | 0.1601 | | | |
| ΔAlb | Decreased | 2.3 | 1.35–3.82 | 0.0020 | 1.9 | 1.11–3.16 | 0.0186 | 2.5 | 1.41–4.30 | 0.0015 | 2.0 | 1.15–3.52 | 0.0145 |
| preAlb | Low | 1.6 | 1.00–2.46 | 0.0490 | | | | 1.7 | 1.03–2.70 | 0.0371 | | | |

HR, hazard ratio; CI, confidence interval; pT, pathological T status; ΔAlb, postoperative decrease in serum albumin; preAlb, preoperative serum albumin levels.

simple blood testing, suggesting that this may be a suitable marker for monitoring the nutritional condition of patients.

In the cancer-bearing state, it is well known that albumin consumption by cancer cells causes albumin depletion [26]. Therefore, when considering the significance of ΔAlb, recurrence within 1 year must be a confounding factor in the association between ΔAlb and the outcomes of NSCLC. To eliminate the influence of recurrence within 1 year, this study excluded patients who recurred within 1 year; then, the prognostic impact of ΔAlb seemed to be independent from recurrence within 1 year. However, in patients with early-stage NSCLC who underwent complete resection, postoperative albumin depletion might reflect a potential cancer recurrence before an apparent cancer recurrence is identified by tests such as computed tomography and tumor markers. Several studies to detect circulating cancer cells [27–30] and circulating tumor DNA [31–33] in blood are underway to investigate potential cancer existence and metastasis in NSCLC. Our results might be supportive to these studies, and it seems useful to investigate changes in serum albumin along with techniques for detection of micro cancer cells. Although adjuvant chemotherapy after complete resection is not currently recommended for pStage IA NSCLC, future improvements in techniques for detecting micro cancer cells, including decreased albumin, might allow the selection of cases that would benefit from adjuvant chemotherapy.

Our combined survival analysis of preAlb and ΔAlb clarified the clinical significance of ΔAlb in terms of the outcomes of patients with NSCLC. In particular, the prognosis of patients in the Alb-Low-Stable group was not significantly worse compared with those of the Alb-High-Decreased and the Alb-High-Stable groups. This result suggested that, even if the preoperative nutritional status was poor, the prognosis of patients with a maintained nutritional

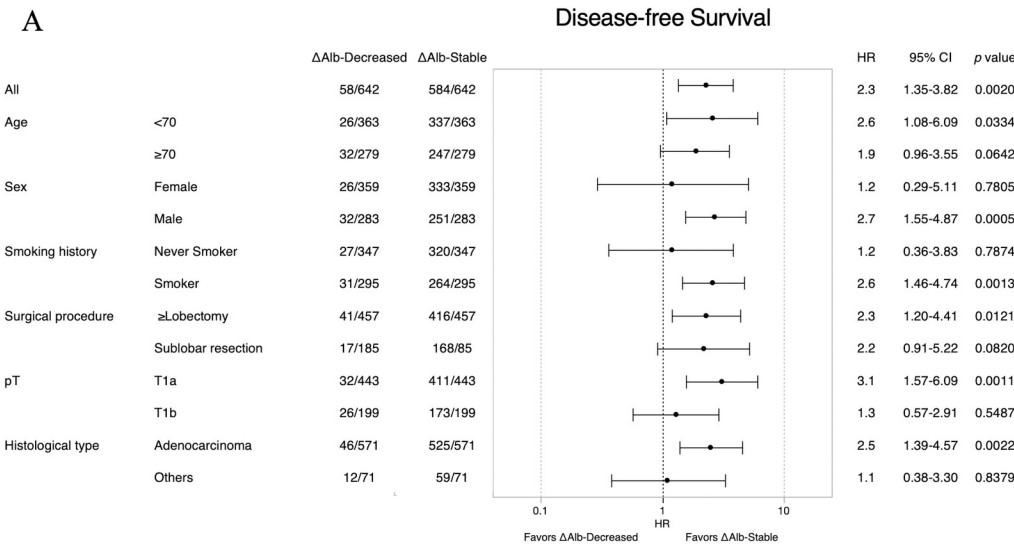

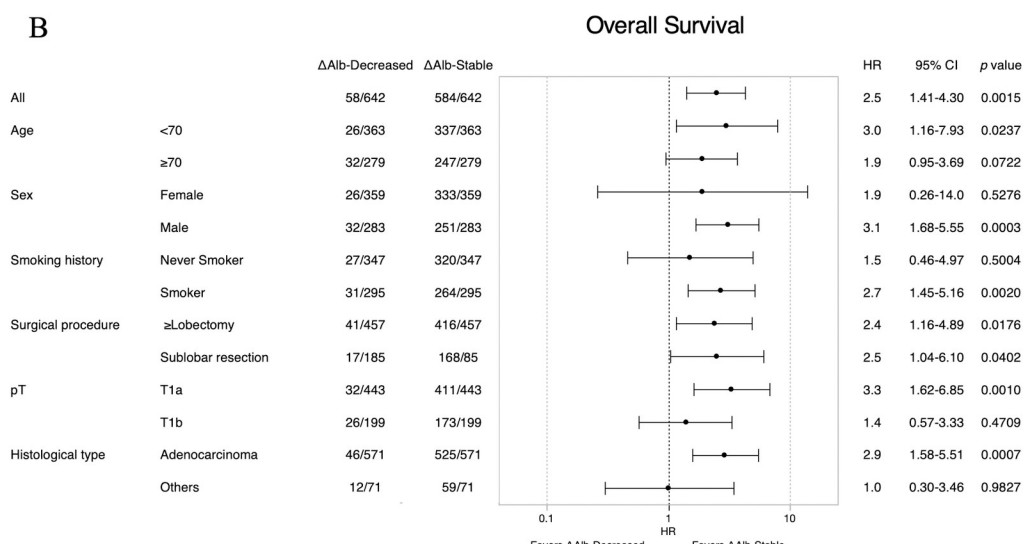

**Fig 4. Forest plot of subgroup analysis for postoperative decrease in serum albumin (ΔAlb) in the validation cohort (VC).** Hazard ratios of ΔAlb for (A) disease-free survival and (B) overall survival in the VC. HR: Hazard ratio, CI: Confidence interval.

status was favorable. From another point of view, even in the preAlb-High group, the prognosis of the Alb-High-Decreased group, in which serum albumin levels were decreased after surgery, was worse than that of the Alb-High-Stable group, and equal to or worse than that of the Alb-Low-Stable group.

Several drugs, including megestrol acetate, ghrelin agonists, and anti-myostatin peptides, are expected to improve cancer cachexia, in addition to nutritional support and physical exercise [34–37]. Recently, anamorelin, which is an orally active ghrelin receptor agonist, has been shown the usefulness for improving cancer cachexia [38–40]. Although there has been no direct evaluation of serum albumin levels, blood sampling data on nutrition, such as prealbumin, have been shown to improve with oral anamorelin [38], and we expect that serum albumin levels may also improve by anamorelin. In addition, in the perioperative period of

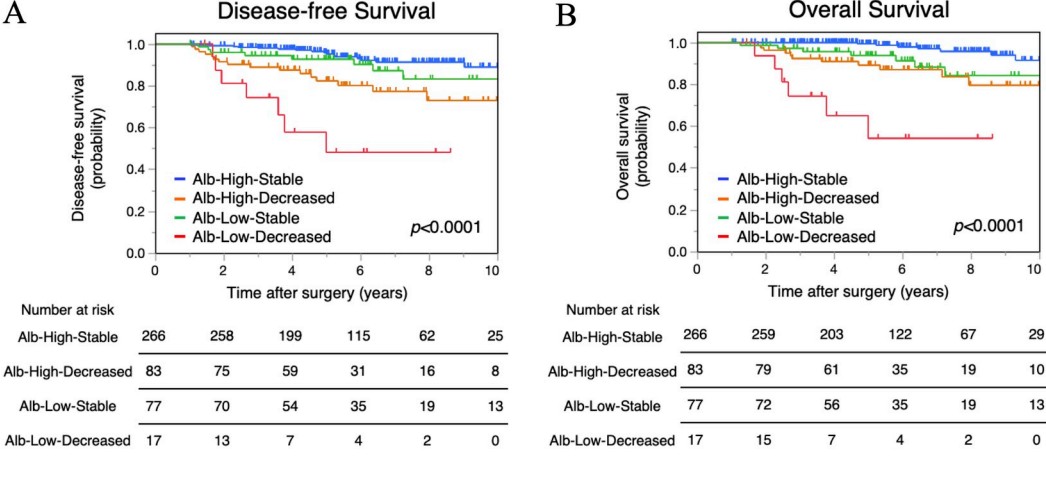

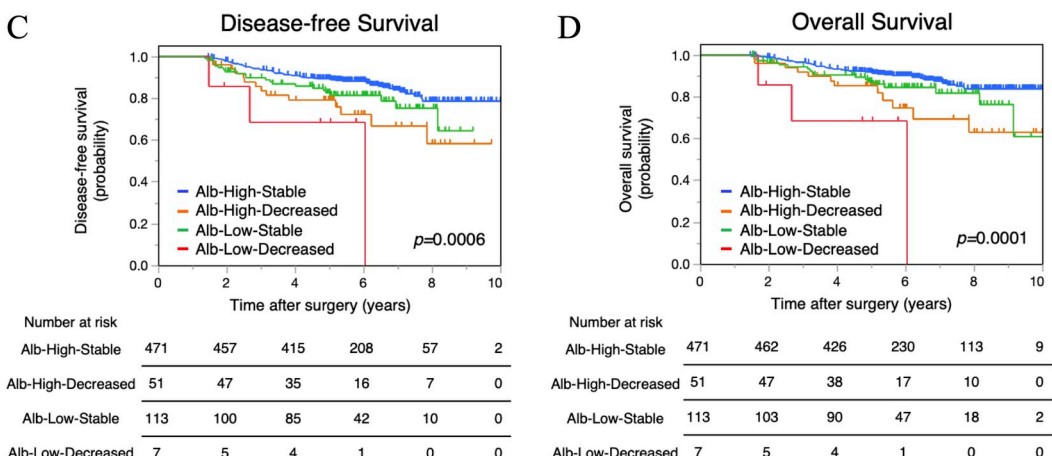

**Fig 5. Combined survival analysis of preoperative serum albumin levels (preAlb) and postoperative decrease in serum albumin (ΔAlb).** Disease-free survival and overall survival of the Alb-Low-Decreased, Alb-Low-Stable, Alb-High-Decreased, and Alb-High-Stable groups in the training cohort (A, B) and in the validation cohort (C, D).

NSCLC, enteral nutrition in addition to active rehabilitation has been shown to improve albumin levels [41], and a randomized controlled trial has been conducted to determine the changes in nutritional status when enteral nutrition is added to perioperative rehabilitation of NSCLC [42]. Therefore, we consider that these strategies might be applied to NSCLC patients after surgery to maintain their nutritional status, leading to improved outcomes.

This study had several limitations. This was a retrospective study; however, the study cohort was relatively large, and 1085 patients were enrolled. We excluded patients who recurred or died within 1 year, and the prognostic impact of preAlb was therefore probably underestimated. In addition, we analyzed ΔAlb at 1 year after surgery in this study, and ΔAlb at 1 year was significantly associated with the subsequent prognosis of early-stage NSCLC; however, to further clarify how ΔAlb is related to the recurrence and prognosis of NSCLC, more detailed examinations of changes in serum albumin over time using serum albumin levels at other time points, such as 1 month, 3 months, and 6 months after surgery, may be required. Furthermore, serum albumin can be influenced by various diseases as well as cancer, such as heart, liver, and renal disease, and care is therefore needed in interpreting the meaning of serum albumin levels in clinical practice. In addition, postoperative smoking status might be one of the related

factors with the postoperative nutritional changes of the patients; however, we have not been able to collect the data on postoperative smoking status and evaluate the influence of postoperative smoking status in this study. There were several limitations in pathological information, such as the TNM classification and adenocarcinoma subtypes; then, it seems necessary to perform the analysis using 8th edition or detailed adenocarcinoma subtypes in the future.

In conclusion, the results of this study demonstrated that ΔAlb was significantly associated with the prognosis of early-stage NSCLC. Therefore, postoperative changes in nutritional status might be important for the outcomes of NSCLC.

## Supporting information

**S1 Fig. The histograms of distributions of preoperative serum albumin levels (preAlb) and postoperative decrease in serum albumin (ΔAlb).** The histograms of distributions of preAlb and ΔAlb in the training cohort (A, B) and in the validation cohort (C, D).
(TIF)

**S2 Fig. Receiver operating characteristic curve (ROC) of training cohort (TC).** ROC curves for preoperative serum albumin levels (preAlb) in the TC. AUC: area under the curve.
(TIF)

**S3 Fig. Survival analysis according to preoperative serum albumin levels (preAlb).** Disease-free survival and overall survival of the preAlb-Low and preAlb-High groups in the training cohort (A, B) and in the validation cohort (C, D).
(TIF)

**S4 Fig. Forest plot of subgroup analysis for postoperative decrease in serum albumin (ΔAlb) in the training cohort (TC).** Hazard ratios of ΔAlb for (A) disease-free survival and (B) overall survival in the TC. HR: hazard ratio, CI: confidence interval.
(TIF)

**S1 Table. Clinicopathological characteristics of patients with pStage IA non-small cell lung carcinoma.**
(DOCX)

**S2 Table. Associations of clinicopathological characteristics with preAlb.**
(DOCX)

**S3 Table. Univariable and multivariable analyses of disease-free survival and overall survival in the training cohort.**
(DOCX)

## Acknowledgments

We thank Takashi Kinoshita from the Medical Information Center, Kyushu University Hospital, for invaluable help with the data collection. We also thank H. Nikki March, PhD, and Susan Furness, PhD, from Edanz Group (https://en-author-services.edanz.com/ac) for editing a draft of this manuscript.

## Author Contributions

**Conceptualization:** Fumihiko Kinoshita, Tetsuzo Tagawa, Takanori Yamashita.

**Data curation:** Fumihiko Kinoshita, Tetsuzo Tagawa, Takanori Yamashita, Taichi Matsubara, Gouji Toyokawa, Kazuki Takada, Taro Oba, Atsushi Osoegawa.

**Formal analysis:** Fumihiko Kinoshita, Tetsuzo Tagawa, Tomoyoshi Takenaka, Kazuki Takada, Taro Oba, Atsushi Osoegawa, Mototsugu Shimokawa.

**Investigation:** Fumihiko Kinoshita, Taichi Matsubara, Gouji Toyokawa, Mototsugu Shimokawa.

**Methodology:** Fumihiko Kinoshita, Takanori Yamashita, Kazuki Takada, Mototsugu Shimokawa.

**Project administration:** Tetsuzo Tagawa, Tomoyoshi Takenaka.

**Supervision:** Tetsuzo Tagawa, Tomoyoshi Takenaka, Koji Yamazaki, Mitsuhiro Takenoyama, Mototsugu Shimokawa, Naoki Nakashima, Masaki Mori.

**Validation:** Tetsuzo Tagawa, Tomoyoshi Takenaka, Mototsugu Shimokawa.

**Visualization:** Tetsuzo Tagawa.

**Writing – original draft:** Fumihiko Kinoshita.

**Writing – review & editing:** Tetsuzo Tagawa, Naoki Nakashima, Masaki Mori.

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
