## [Decision Letter · Decision Letter 0]

25 May 2021

PONE-D-21-10743

Prognostic value of postoperative decrease in serum albumin on surgically resected early-stage non-small cell lung carcinoma: a multicenter retrospective study

PLOS ONE

Dear Dr. Tagawa,

Thank you for submitting your manuscript to PLOS ONE. After careful consideration, we feel that it has merit but does not fully meet PLOS ONE’s publication criteria as it currently stands. Therefore, we invite you to submit a revised version of the manuscript that addresses the points raised during the review process.

Reviewers raised constructive comments on your manuscript. They pointed out the clarification of albumin measurement across multiple cohorts. And, you need to more clearly describe study design and statistical methods with updated staging system.

We look forward to receiving your revised manuscript.

Kind regards,

Hyun-Sung Lee, M.D., Ph.D.

Academic Editor

PLOS ONE

Journal Requirements:

2.  Thank you for including your ethics statement: "This study was reviewed and approved by our institutional review boards (Kyushu University Hospital [KUH], IRB No. 2019-232; Kyushu Cancer Center [KCC], IRB No. 2019-56; Kyushu Medical Center [KMC], IRB No. 19D152)."

a) Please provide additional details regarding participant consent. In the ethics statement in the Methods and online submission information, please ensure that you have specified (1) whether consent was informed and (2) what type you obtained (for instance, written or verbal, and if verbal, how it was documented and witnessed). If your study included minors, state whether you obtained consent from parents or guardians. If the need for consent was waived by the ethics committee, please include this information.

'The funders had no role in study design, data collection and analysis, decision to publish, or preparation of the manuscript.'

Reviewers' comments:

Reviewer's Responses to Questions

**Comments to the Author**

1. Is the manuscript technically sound, and do the data support the conclusions?

Reviewer #1: Yes

Reviewer #2: Yes

2. Has the statistical analysis been performed appropriately and rigorously? 

Reviewer #1: Yes

Reviewer #2: No

3. Have the authors made all data underlying the findings in their manuscript fully available?

Reviewer #1: Yes

Reviewer #2: Yes

4. Is the manuscript presented in an intelligible fashion and written in standard English?

Reviewer #1: Yes

Reviewer #2: Yes

5. Review Comments to the Author

Reviewer #1: Authors investigated the influence of the postoperative changes in the serum albumin level on the postoperative prognosis of the patients with Stage IA non-small cell lung cancer who underwent complete resection. Through the study, they found that the decrease of serum albumin level had a negative impact on the postoperative progression-free survival and overall survival. They set a cut-off level of the degree of changes in serum albumin levels from the training set, and adopt it to the validation set. The methodological design for the study was well organized, and the statistical analyses were well performed. There are some issues to settle.

1. Abstract, Conclusions: The first sentence is a bit too abstracted and obscure. It should be more concrete for summarizing the study. It may be “We demonstrated a negative impact of postoperative decrease of the serum albumin on the prognosis of patients with early-stage NSCLC.”

2. Patients and Methods, Study patients: The flow of the patients included in the study was a bit too complicated to understand by reading the text. Please provide a CONSORT diagram as a Figure for readers to understand the whole schema of the participants.

3. Patients and Methods, Evaluation of serum albumin: the data of albumin level was collected from three different hospitals. Are there any difference in the measuring method for serum albumin?

4. To be related with the #3, is there any adjustment with the serum calcium level to determine the serum albumin level?

5. Table 1: Does the smoking refer the preoperative status? I think the postoperative smoking status is more related with the postoperative changes of the patients. Could you provide the postoperative smoking status, if possible?

6: Authors did not include any pulmonary status, such as COPD or restrictive pulmonary disease. I wonder the nutritional status may influenced by those types of pulmonary diseases especially after the lung resections. Could you incorporate the pulmonary status in the statistical models for prognosis?

Reviewer #2: This is a multi-institution, retrospective cohort study that examines 1,085 patients with a history of pathologic stage IA non-small cell lung cancer (NSCLC) in 2003 and 2014 designed to analyze the clinical significance of a postoperative change of serum albumin at 1 year after surgery as a marker of nutritional status on the long-term outcomes. You have concluded the alteration of serum albumin between preoperative and postoperative 1-year levels is an independent prognostic factor for disease-free survival (DFS) and overall survival (OS) in 443 patients in a training cohort and 642 patients in validation cohorts with pstage IA NSCLC.

1. Through large multi-institutional cohorts of patients with pstage IA NSCLC, the authors attempted to determine the association of nutritional change in albumin with long-term survival. However, this manuscript has several limitations to connect postoperative parameters, which may be affected by many confound variables including minimally invasive surgery, dietary modification, and additional treatment, with long term outcomes.

2. Based on your Figure 3, the effect of albumin changes looks more significant in T1a (based on the 7th edition) and adenocarcinoma. Subgroup analysis of T1a lung adenocarcinoma with the updated staging system (8th edition), which can be divided into new T1a and T1b, may be more interesting. Furthermore, more detailed histologic or pathologic findings in adenocarcinoma may be considered.

3. Please describe the method to measure serum albumin. And, please clarify if the same methods are applied to multiple cohorts and the range or standard deviations are similar across the cohorts.

The cut off values of albumin are set up based on OS, which will lead to distinct survival curve of at least a training cohort. More objective way to set up the cutoff value may be required. Based on your finding considering both pre-albumin and albumin change, only 3.8 (17/443) % in a training cohort and 1% (7/642) in a validation cohort are classified into albumin-low-decreased, which has the worst prognosis. These numbers look too small to generalize your findings. And, the albumin values do not outperform other variables such as old age and male. There is no comparison of albumin changes with other nutritional markers.

4. If you consider that albumin change and prealbumin are modifiable variables, please describe more detailed strategies to improve the nutritional status.

5. As I know, adjuvant tegafur/uracil (UFT) chemotherapy is recommended for patients with completely resected Stage I NSCLC in Japan. I wonder if your cohorts include patients with adjuvant Tx.

6. Multivariate is an incorrect term since you have used a dependent variable such as OS or DFS. Multivariable analysis would be a correct terminology. And, you have not described how you have selected variables for multivariable analysis. T staging and smoking status are excluded although they are statistically significant during the univariable analysis.

6. PLOS authors have the option to publish the peer review history of their article (what does this mean?). If published, this will include your full peer review and any attached files.

Reviewer #1: No

Reviewer #2: No

---

## [Author Response · Author response to Decision Letter 0]

23 Jul 2021

Responses to Reviewers

Reviewer #1

Authors investigated the influence of the postoperative changes in the serum albumin level on the postoperative prognosis of the patients with Stage IA non-small cell lung cancer who underwent complete resection. Through the study, they found that the decrease of serum albumin level had a negative impact on the postoperative progression-free survival and overall survival. They set a cut-off level of the degree of changes in serum albumin levels from the training set, and adopt it to the validation set. The methodological design for the study was well organized, and the statistical analyses were well performed. There are some issues to settle.

Comment 1: Abstract, Conclusions: The first sentence is a bit too abstracted and obscure. It should be more concrete for summarizing the study. It may be “We demonstrated a negative impact of postoperative decrease of the serum albumin on the prognosis of patients with early-stage NSCLC.”

Answer 1: Thank you very much for your comments on our manuscript. As you pointed out, we have corrected the sentence at Conclusions of Abstract, to summarize the research more concretely.

Change in manuscript 1:

(page 2, line 42-44)

We demonstrated a negative impact of postoperative decrease of the serum albumin on the prognosis of patients with early-stage NSCLC. 

Comment 2: Patients and Methods, Study patients: The flow of the patients included in the study was a bit too complicated to understand by reading the text. Please provide a CONSORT diagram as a Figure for readers to understand the whole schema of the participants.

Answer 2: 

Thank you very much for your very useful advice. We have created an additional Figure with the consort diagram. 

Change in manuscript 2:

(page 4, lines 78-79).

The consort diagram of patient selection was shown in Figure 1 .

(page 4, lines 87-88).

Figure 1. Consort diagram of patient selection. pStage, pathological stage; NSCLC, non-small cell lung carcinoma.

(Figure 1)

Comment 3: Patients and Methods, Evaluation of serum albumin: the data of albumin level was collected from three different hospitals. Are there any difference in the measuring method for serum albumin?

Answer 3: 

For measuring serum albumin levels, all of three institutions used the modified bromocresol purple method. 

Change in manuscript 3:

(page 5, lines 92-93)

For measuring serum albumin levels, all of three institutions used the modified bromocresol purple method.

Comment 4: 

To be related with the #3, is there any adjustment with the serum calcium level to determine the serum albumin level?

Answer 4:

As you mentioned, most of the serum calcium is bound to serum albumin, and there is a close relationship between the both of them. However, as far as we know, although it is important to calculate adjusted calcium levels, in hypoalbuminemia, to determine clinically important ionized calcium levels, there might be no need to adjust albumin levels with calcium levels.

Change in manuscript 4:

There is no change in manuscript.

Comment 5: Table 1: Does the smoking refer the preoperative status? I think the postoperative smoking status is more related with the postoperative changes of the patients. Could you provide the postoperative smoking status, if possible?

Answer 5:.

We are very sorry, but we have not been able to collect the data on postoperative smoking status, and it would be difficult to evaluate it in this study. We added a statement to Limitation regarding the lack of data on postoperative smoking status.

Change in manuscript 5:

(page 13, lines 253-256)

In addition, postoperative smoking status might be one of the related factors with the postoperative nutritional changes of the patients; however, we have not been able to collect the data on postoperative smoking status and evaluate the influence of postoperative smoking status in this study.

Comment 6: Authors did not include any pulmonary status, such as COPD or restrictive pulmonary disease. I wonder the nutritional status may influenced by those types of pulmonary diseases especially after the lung resections. Could you incorporate the pulmonary status in the statistical models for prognosis?

Answer 6:.

We added the presence of pulmonary comorbidities, such as IP, COPD, and asthma, to our analysis. As the result, we found that the presence of pulmonary comorbidities was an important prognostic factor, but it did not show the associations with preAlb and ΔAlb.

Change in manuscript 6:

(Table1, Table2, Supplementary Table 1, Supplementary Table 2, Supplementary Table 3)

We added the presence of pulmonary comorbidities to the analyses.

(page 4, lines 81-84)

Clinicopathological characteristics included age, sex, smoking history, pulmonary comorbidities, surgical procedure, pathological T status (pT), histological type, vascular invasion, and lymphatic invasion. Pulmonary comorbidities included interstitial pneumonia, chronic obstructive pulmonary disease, or asthma. 

(page 9, lines 167-171)

In the VC, multivariable analysis identified older age (p<0.0001), male sex (p<0.0001), pulmonary comorbidity (p=0.0017), and ΔAlb-Decreased (p=0.0186) as independent prognostic factors for DFS, and older age (p<0.0001), male sex (p=0.0001), pulmonary comorbidity (p=0.0006), lobectomy (p=0.0247), and ΔAlb-Decreased (p=0.0145) as independent prognostic factors for OS (Table 2).

 

Reviewer #2: 

This is a multi-institution, retrospective cohort study that examines 1,085 patients with a history of pathologic stage IA non-small cell lung cancer (NSCLC) in 2003 and 2014 designed to analyze the clinical significance of a postoperative change of serum albumin at 1 year after surgery as a marker of nutritional status on the long-term outcomes. You have concluded the alteration of serum albumin between preoperative and postoperative 1-year levels is an independent prognostic factor for disease-free survival (DFS) and overall survival (OS) in 443 patients in a training cohort and 642 patients in validation cohorts with pstage IA NSCLC.

Comment 1: Through large multi-institutional cohorts of patients with pstage IA NSCLC, the authors attempted to determine the association of nutritional change in albumin with long-term survival. However, this manuscript has several limitations to connect postoperative parameters, which may be affected by many confound variables including minimally invasive surgery, dietary modification, and additional treatment, with long term outcomes.

Answer 1: Thank you very much for your comments on our manuscript. As you pointed, it is true that postoperative serum albumin levels are influenced by many factors, such as surgical procedures, dietary modifications, and additional treatments. However, as we mentioned as the purpose of the study (page 3, line 60-62), we consider that serum albumin has significance as a nutritional marker for comprehensive and simple evaluation of these factors that affect postoperative nutritional status. 

Change in manuscript 1:

There is no change in manuscript.

Comment 2: Based on your Figure 3, the effect of albumin changes looks more significant in T1a (based on the 7th edition) and adenocarcinoma. Subgroup analysis of T1a lung adenocarcinoma with the updated staging system (8th edition), which can be divided into new T1a and T1b, may be more interesting. Furthermore, more detailed histologic or pathologic findings in adenocarcinoma may be considered.

Answer 2:　

Thank you for your very constructive feedback. However, we are very sorry but, in this retrospective study, we have not been able to collect the data on pathological invasive area, and it would be difficult to update staging system to 8th edition in this study. 

In terms of histological subtypes of lung adenocarcinoma, we performed a subgroup analysis in the training cohort. The results of the subgroup analysis showed that ΔAlb-Decreased patients tended to have a poor prognosis in both lepidic predominant adenocarcinoma and non-lepidic predominant adenocarcinoma patients. However, there were missing data on histological subtypes of lung adenocarcinoma in the validation cohort; then, subgroup analysis could not be performed in the validation cohort.

We added the sentence bout the TNM 8th edition and histological subtypes in Limitation.

Change in manuscript 2:

(page 14, line 270-273).

There were several limitations in pathological information, such as the TNM classification and adenocarcinoma subtypes; then, it seems necessary to perform the analysis using 8th edition or detailed adenocarcinoma subtypes in the future.

Comment 3: Please describe the method to measure serum albumin. And, please clarify if the same methods are applied to multiple cohorts and the range or standard deviations are similar across the cohorts.

The cut off values of albumin are set up based on OS, which will lead to distinct survival curve of at least a training cohort. More objective way to set up the cutoff value may be required. Based on your finding considering both pre-albumin and albumin change, only 3.8 (17/443) % in a training cohort and 1% (7/642) in a validation cohort are classified into albumin-low-decreased, which has the worst prognosis. These numbers look too small to generalize your findings. And, the albumin values do not outperform other variables such as old age and male. There is no comparison of albumin changes with other nutritional markers.

Answer 3: 

Thank you for your important advice on the fundamentals of this study. For measuring serum albumin levels, all of three institutions use the modified bromocresol purple method. The median preAlb and ΔAlb in the TC were 4.3 g/dL (range, 3.1 to 5.3 g/dL) and 4.8 % (range, -29.2 to 79.2 %), respectively. In the VC, the median preAlb and ΔAlb in the TC were 4.3 g/dL (range, 3.0 to 5.4 g/dL) and 2.4 % (range, -30.0 to 43.9 %), respectively. The histograms of distributions of preAlb and ΔAlb in each cohort were shown in Supplementary Figure 1. We thought the distributions of preAlb and ΔAlb were similar in the TC and the VC.

In this study, the cutoff value of ΔAlb was set in the training cohort and validated in the validation cohort; moreover, multivariable analysis and subgroup analysis in the validation cohort showed that ΔAlb was an independent factor in both DFS and OS. Therefore, we consider that we have sufficiently examined the cutoff of ΔAlb. Then, if we will further examine the cutoff, it may need to be a prospective study in larger cohort.

What we want to elucidate in this study is the importance of ΔAlb-Decreased, including Alb-High-Decreased group. Then, we think that the numbers of each groups in the additional 4-group analysis dose not largely affect the hypothesis of this study. We added the sentence that “even in the preAlb-High group, the prognosis of the Alb-High-Decreased group, in which serum albumin levels were decreased after surgery, was worse than that of the Alb-High-Stable group, and equal to or worse than that of the Alb-Low-Stable group.” in the manuscript to clarify our meaning point.

The results of multivariable and subgroup analyses have shown that other factors, such as age and sex, and ΔAlb are both independently important factors, and we don't think it is necessary to argue about which one is superior to the other factors.

We would like to compare serum albumin with other nutritional indices in the future. However, as we mentioned in Answer 1, we consider that serum albumin is very usefulness as a simple and comprehensive nutritional marker.

Change in manuscript 3:

(page 5, lines 102-104)

For measuring serum albumin levels, all of three institutions use the modified bromocresol purple method. 

(page 6, lines 117-120)

In terms of serum albumin, the median preAlb and ΔAlb in the TC were 4.3 g/dL (range, 3.1 to 5.3 g/dL) and 4.8 % (range, -29.2 to 79.2 %), respectively. In the VC, the median preAlb and ΔAlb in the TC were 4.3 g/dL (range, 3.0 to 5.4 g/dL) and 2.4 % (range, -30.0 to 43.9 %), respectively. The histograms of distributions of preAlb and ΔAlb in each cohort were shown in Supplementary Figure 1

(page 13, lines 244-247)

From another point of view, even in the preAlb-High group, the prognosis of the Alb-High-Decreased group, in which serum albumin levels were decreased after surgery, was worse than that of the Alb-High-Stable group, and equal to or worse than that of the Alb-Low-Stable group. 

(Supplementary Figure 1)

Supplementary Figure 1. The histograms of distributions of preoperative serum albumin levels (preAlb) and postoperative decrease in serum albumin (ΔAlb). The histograms of distributions of preAlb and ΔAlb in the training cohort (A, B) and in the validation cohort (C, D).

Comment 4: If you consider that albumin change and prealbumin are modifiable variables, please describe more detailed strategies to improve the nutritional status.

Answer 4:　

Recently, anamorelin, which is an orally active ghrelin receptor agonist, has been shown the usefulness for improving cancer cachexia. The studies supporting the usefulness of anamorelin have been described in detail. Although there has been no direct evaluation of serum albumin levels, blood sampling data on nutrition, such as prealbumin, have been shown to improve with oral anamorelin, and we expect that serum albumin levels may also improve by anamorelin.

In addition, in the perioperative period of non-small cell lung cancer, enteral nutrition in addition to active rehabilitation has been shown to improve albumin levels, and there is a possibility that enteral nutrition may also be useful in improving albumin levels.

Change in manuscript 4:

(page 13, lines 249-256)

Recently, anamorelin, which is an orally active ghrelin receptor agonist, has been shown the usefulness for improving cancer cachexia.38-40 Although there has been no direct evaluation of serum albumin levels, blood sampling data on nutrition, such as prealbumin, have been shown to improve with oral anamorelin,38 and we expect that serum albumin levels may also improve by anamorelin. In addition, in the perioperative period of NSCLC, enteral nutrition in addition to active rehabilitation has been shown to improve albumin levels,41 and a randomized controlled trial has been conducted to determine the changes in nutritional status when enteral nutrition is added to perioperative rehabilitation of NSCLC.42

(Reference)

38. Katakami N, Uchino J, Yokoyama T, et al. (2018) Anamorelin (ONO-7643) for the treatment of patients with non-small cell lung cancer and cachexia: Results from a randomized, double-blind, placebo-controlled, multicenter study of Japanese patients (ONO-7643-04). Cancer. 124:606-616. 

39. Takayama K, Katakami N, Yokoyama T, et al. (2016) Anamorelin (ONO-7643) in Japanese patients with non-small cell lung cancer and cachexia: results of a randomized phase 2 trial. Support Care Cancer. 24:3495-505. 

40. Temel JS, Abernethy AP, Currow DC, et al. (2016) Anamorelin in patients with non-small-cell lung cancer and cachexia (ROMANA 1 and ROMANA 2): results from two randomised, double-blind, phase 3 trials. Lancet Oncol. 17:519-531. 

41. Ding Q, Chen W, Gu Y, et al. (2020) Accelerated rehabilitation combined with enteral nutrition in the management of lung cancer surgery patients. Asia Pac J Clin Nutr. 29:274-279. 

42. Ji X, Ding H. (2020) The efficacy of enteral nutrition combined with accelerated rehabilitation in non-small cell lung cancer surgery: A randomized controlled trial protocol. Medicine (Baltimore). 99:e23382. 

Comment 5: As I know, adjuvant tegafur/uracil (UFT) chemotherapy is recommended for patients with completely resected Stage I NSCLC in Japan. I wonder if your cohorts include patients with adjuvant Tx.

Answer 5:.

As you mentioned, we also agree that adjuvant chemotherapy may have a significant effect on the postoperative albumin changes. Therefore, we excluded patients who had received adjuvant chemotherapy including UFT in this study. We are sorry for not mentioning the exclusion of patients who received postoperative adjuvant chemotherapy. The information about the exclusion of patients who underwent postoperative adjuvant chemotherapy has been added.

Change in manuscript 5:

(page 4, lines 73-75)

Patients who received chemotherapy within 1 year after surgery including adjuvant chemotherapy were also excluded from this study.

Comment 6: Multivariate is an incorrect term since you have used a dependent variable such as OS or DFS. Multivariable analysis would be a correct terminology. And, you have not described how you have selected variables for multivariable analysis. T staging and smoking status are excluded although they are statistically significant during the univariable analysis. 

Answer 6:.

Thank you for pointing out the error in use of the words, and “Univariate” and “Multivariate” have been corrected to “Univariable” and “Multivariable”.

As for multivariable analysis, after applying multivariable analysis to all factors used in univariable analysis at the same time, we eliminated less important factors with high p-values one by one and reduced factors so that only factors with p<0.05 were included. pT status and smoking history were also excluded during the factor reduction phase. A description of the multivariable analysis method has been added to the Method.

Change in manuscript 6:

(page 5, lines 102-104)

As for multivariable analysis, after applying multivariable analysis to all factors used in univariable analysis at the same time, we eliminated less important factors with high p-values one by one and reduced factors so that only factors with p<0.05 were included.

---

## [Editor Report · Decision Letter 1]

18 Aug 2021

Prognostic value of postoperative decrease in serum albumin on surgically resected early-stage non-small cell lung carcinoma: a multicenter retrospective study

PONE-D-21-10743R1

Dear Dr. Tagawa,

We’re pleased to inform you that your manuscript has been judged scientifically suitable for publication and will be formally accepted for publication once it meets all outstanding technical requirements.

Kind regards,

Hyun-Sung Lee, M.D., Ph.D.

Academic Editor

PLOS ONE
---

## [Editor Report · Acceptance letter]

25 Aug 2021

PONE-D-21-10743R1 

Prognostic value of postoperative decrease in serum albumin on surgically resected early-stage non-small cell lung carcinoma: a multicenter retrospective study 

Dear Dr. Tagawa:

I'm pleased to inform you that your manuscript has been deemed suitable for publication in PLOS ONE. Congratulations! Your manuscript is now with our production department. 

Kind regards, 

on behalf of

Dr. Hyun-Sung Lee 

Academic Editor

PLOS ONE